# Short communication: miRNA122 interrogation via PCR-Free method to track liver recovery

Antonio Marín-Romero[1][☺], Daniel E. Di Zeo-Sánchez[2,3☺], Mavys Tabraue-Chávez[1], Marina Villanueva-Paz[2,3], Jose M. Pinazo-Bandera[2,3], Judith Sanabria-Cabrera[2,3], Miren García-Cortés[2,3], Juan J. Díaz-Mochón[4,5,6,7‡], M. Isabel Lucena[2,3*], Raúl J. Andrade[2,3‡], Camilla Stephens[2,3‡], Salvatore Pernagallo[1‡*]

**1** DESTINA Genomica S.L., Edificio BIC, Parque Tecnológico Ciencias de la Salud, Granada, Spain, **2** Servicios de Aparato Digestivo y Farmacología Clínica, Instituto de Investigación Biomédica de Málaga y Plataforma en Nanomedicina-IBIMA Plataforma, Hospital Universitario Virgen de la Victoria, Universidad de Málaga, Málaga, Spain, **3** Centro de Investigación Biomédica en Red Enfermedades Hepáticas y Digestivas, Madrid, Spain, **4** Department of Medicinal & Organic Chemistry, Faculty of Pharmacy, University of Granada, Granada, Spain, **5** Centre for Genomics and Oncological Research, Pfizer/University of Granada/Andalusian Regional Government, Parque Tecnológico Ciencias de la Salud, Granada, Spain, **6** Unit of Excellence in Chemistry Applied to Biomedicine and the Environment of the University of Granada, Granada, Spain, **7** Instituto de Investigación Biosanitaria, Granada, Spain

☺ These authors contributed equally to this work.
‡ JJD-M, RJA, CS and SP shared senior authorship on this work.
* salvatore@destinagenomics.com (SP); mlucena@uma.es (MIL)

## Abstract

There is currently a need to investigate new biomarkers of acute liver injury (ALI) that are highly specific to the liver and capable of detecting early-stage liver damage. In this regard, circulating microRNAs (miRNAs), particularly microRNA122 (miRNA122), have recently been proposed as promising new candidate biomarkers. However, the quantification of miRNAs is not a standardized technique and presents several challenges that hinder its routine use. The aim of this work is to validate the innovative Dynamic Chemical Labeling (DCL) PCR-Free technology for its ability to detect miRNA122 in patient samples using Luminex xMAP platforms. The DCL PCR-Free technology was used to directly measure and monitor miRNA122 levels in serum samples from patients with ALI. Patients were monitored throughout the recovery process from liver injury, from the time of detection and for up to 30 days afterwards, with follow-up over three separate visits. The results demonstrate the robustness of the method, with sensitivity of 3.36% and precision of 99.80%, and show a high correlation between miRNA122 and traditional liver injury markers such as ALT (r=0.8150, p=0.0001), AST (r=0.7895, p=0.0002) and TBL (r=0.2646, p=0.3406) throughout the ALI recovery process. In conclusion, measuring miRNA122 levels using the DCL PCR-Free method provides a distinctive approach, not only for diagnosing ALI patients but also for effectively monitoring disease progression, tracking liver recovery, and evaluating treatment effectiveness.

**Data availability statement:** All relevant data are within the paper and its Supporting information files.

**Funding:** This work was supported by DESTINA Genomica S.L. through the Eurostars-2 grant, which is co-funded by the EUREKA member countries (CDTI in Spain) and the European Union's Horizon 2020 Framework Programme (project code: E! 114589 - LiverAce). As part of this project, DESTINA Genomica S.L. provided salaries and research materials for the authors (AMR, MTC and SP) and research materials for JJD-M. This work was also funded by the grant from the Instituto de Salud Carlos III, which is co-funded by the European Union and the Fondo Europeo de Desarrollo Regional (FEDER) (contract numbers: PI21/01248 and PI19-00883). MVP was supported by a Sara Borrell research contract (CD21/00198) from the ISCIII and the Consejería de Salud de Andalucía. DDS holds a Doctorados IIS-empresa en Ciencias y Tecnologías de la Salud (i-PFIS) research contract (IFI21/00034) from ISCIII. CIBERehd is funded by ISCIII. The funders had no role in the design of the study, the collection, analysis or interpretation of the data, the decision to publish, or the preparation of the manuscript. No other external funding was received for this study.

**Competing interests:** AMR, MTC and SP are employed by DESTINA Genomica S.L. Both SP and JJD-M hold shares in DESTINA Genomica S.L. Additionally, JJD-M serves as a board member and director of DESTINA Genomics Ltd., the mother company that owns 100% of DESTINA Genomica S.L. DESTINA Genomica S.L. plans to develop a commercial kit for performing the miRNA122 assay described in this study. This declaration does not affect our adherence to all PLOS ONE policies on data and material sharing, as outlined in the journal's author guidelines.

## Introduction

In clinical practice, most liver diseases are detected by measuring common circulating markers of liver injury. The most common of these biomarkers include the enzyme activities of aminotransferases (alanine aminotransferase [ALT] and aspartate aminotransferase [AST]), alkaline phosphatase (ALP) and γ-glutamyl transferase (GGT), while markers of liver function mainly include total bilirubin (TBL) and prothrombin activity [1]. However, markers such as ALT and AST are not liver-specific, being also found in heart and muscle tissues. This means their levels are influenced by factors other than liver health [2,3]. Therefore, these markers offer limited insight into disease mechanisms and prognosis [4]. Therefore, there is a need to validate new, liver-specific biomarkers that can detect the onset of liver injury at an early stage, allowing for prompt management and prevention of progression.

In the search for new liver-specific biomarkers, the levels of several circulating microRNAs (miRNAs) have been proposed as candidates for various liver diseases, including drug-induced liver injury (DILI) [5], viral hepatitis, metabolic dysfunction-associated steatotic liver disease, and hepatocellular carcinoma [6–11]. Of these,, miRNA-122-5p (miRNA122) has demonstrated high liver specificity and sensitivity for the detection of acute liver injury (ALI) and appears to be less susceptible to damage in other tissues [12,13]. Unlike traditional markers of liver damage, which are thought to increase directly as a result of hepatocyte death and leak into the bloodstream, miRNA122 may also have mechanistic properties, with possible roles in metabolism, inflammation, fibrosis and DILI having been described [14,15]. Based on these properties, miRNA122 has been endorsed by the US Food and Drug Administration as a potential biomarker for liver disease[16].

The detection of miRNAs in clinical samples poses several challenges for its routine implementation. One of the most widely used methods for quantifying nucleic acids is to use polymerases to amplify and detect these molecules by RT-qPCR [17]. However, RT-qPCR is usually time-consuming and requires laborious sample preparation. It is also associated with several technical challenges in miRNA detection. These include degradation of the miRNAs, variability due to the pre-extraction and reverse transcription steps, inhibition of the polymerase, the risk of contamination, and the presence of multiple isoforms with different sequences [18,19].

Various efforts have been undertaken to address these challenges and directly detect miRNAs [20–22]. Over the past decade, our research group has pioneered Dynamic Chemical Labeling (DCL), an advanced PCR-Free method that enables the direct detection of circulating miRNAs in body fluids, eliminating the need for extraction, reverse transcription, and PCR amplification [23–27]. In essence, the DCL PCR-Free method uses a modified peptide nucleic acid (PNA) capture probe containing an abasic site (DGL Probe) and a specially designed biotinylated aldehyde-modified SMART nucleobase (SMART-Base Biotin) to interrogate complementary nucleic acids. The DCL PCR-Free procedure is based on two critical molecular interactions: i) the hybridisation of the DGL Probe with single-stranded miRNA; and (ii) the highly specific reaction of SMART-Base Biotin with the secondary amine found at

the abasic unit of the DGL Probe. This reaction is harnessed by the Watson-Crick base pairing rule to avoid false positives and achieve single-base resolution [28–32] (S1 Fig). This technology enables the absolute quantification of miRNAs on any bead-based immunoassay platform, thereby streamlining the analytical workflow for miRNAs. It also provides a promising avenue for clinical diagnostics, as it allows nucleic acid tests (NATs) to be performed with the simplicity of immunoassays, while maintaining single-nucleotide resolution [33–36].

In this study, we used the DCL PCR-Free method alongside the most widely used bead-based immunoassay platform - Luminex's xMAP technology - to directly measure circulating levels of miRNA122 in patients with ALI. We monitored patients' serum levels of miRNA122 throughout their recovery from the acute episode and compared them with traditional liver markers. This comprehensive evaluation aimed to validate the disease-monitoring capability of miRNA122 specifically, through the application of the unique DCL PCR-Free method. This approach represents the first use of this technology for monitoring the course of acute liver injury.

## Materials and methods

### Study protocol and sample collection

The ALI cases were recruited prospectively from the Virgen de la Victoria University Hospital in Málaga, Spain, between 12/09/2022 and 05/03/2024. The inclusion criteria for ALI at the time of enrolment in the study were as follows: ALT ≥ 5 × the upper limit of normal (ULN), ALT ≥ 3 × ULN + TBL > 2 × ULN, or ALP ≥ 2 × ULN. Serial samples from ALI patients were collected on day 1 (recognition, Visit 1), day 7 (Visit 2) and ≥30 days after recognition of the acute episode (Visit 3) with a clinical blood analysis performed at each time point to monitor liver profile values. All patients had decreased liver profile values at the third time point compared to earlier time points. No additional clinical history data were retrieved for this study.

Tests were performed to diagnose viral hepatitis (hepatitis A, B, C, and E viruses; cytomegalovirus; and Epstein-Barr virus) and autoimmune hepatitis. In cases where a drug was suspected of causing hepatotoxicity, its administration was stopped. The pattern of liver injury (hepatocellular, cholestatic, or mixed) was determined by calculating the ALT-to-ALP ratio (R) using multiples of the ULN from the first available blood analysis after ALI recognition [37]. Severity was assessed using the severity index defined by Aithal et al. [38]. After extraction, blood samples were centrifuged at 2000 x g for 10 minutes, transferred to serum iwithin 1 hour, and stored at −80°C until analysis.

The study protocol, which conforms to the ethical guidelines of the Declaration of Helsinki and was approved by the local Ethics Committee at the Virgen de la Victoria University Hospital in Malaga, Spain (Comité de Ética de la Investigación Provincial de Málaga) on the  30 June 2022. All patients gave written informed consent before being included in the study.

### Quantification of traditional liver profile biomarkers

Quantification of these markers was performed at the Virgen de la Victoria University Hospital in Malaga (Spain), according to the standard methodology applied in clinical practice. In brief, ALT, AST and ALP activities were calculated using luminescence and the Atellica™ CH Alanine Aminotransferase P5P, Atellica™ CH Aspartate Aminotransferase P5P and Atellica™ CH Alkaline Phosphatase assays. TBL was quantified using a chemical oxidation method and the Atellica CH Total Bilirubin_2 kit. All measurements were performed on the Atellica Solution instrument (Siemens).

### Direct quantification of miRNA122

miRNA122 was analyzed in serum samples using the DCL PCR-Free method, a technique that is commercialized by DESTINA Genomica S.L. in Spain as the LiverAce® kit. This assay enables for the direct quantitative analysis of circulating miRNA122 in serum samples using the Luminex MAGPIX platform. The platform is operated using the xPONENT

software. Briefly, a volume of 25 µL of serum was treated with 75 µL of Stabiltech buffer containing 1250 Magplex beads conjugated to DGL-122 probes (referred to as 122 Capture Beads). The samples were then incubated for two hours at 30 ºC with shaking at 800 rpm using an orbital shaker. After incubation, the beads were pelleted and washed three times with 200 µL of PBS containing 0.1% Tween 20. The beads were then incubated for an additional hour of treatment at 40 ºC, while shaking at 800 rpm in a solution of PBS containing 1% BSA and 0.05% NaN$_3$, including 5 µM of SMART-Base Biotin and 1 mM of sodium cyanoborohydride [reducing agent (RA)]. Following a further three additional wash cycles, the beads were incubated for 30 minutes at 30 ºC while shaking at 800 rpm with 50 µL of PBS containing 1% BSA and 0.05 NaN$_3$, supplemented with 2 µg/mL of streptavidin-R-phycoerythrin (SA-PE). After three additional wash cycles, the beads were resuspended in 120 µL of PBS containing 0.1% Tween 20 and analyzed on the MAGPIX platform, using a 100 µL injection volume on the MAGPIX platform. Incubations were performed using a 96-well plate shaker (VWR° Microplate Shaker). Washings were performed using a Biotek 405 TS semi-automatic washer.

To quantify sample levels, a seven-point calibration curve was performed in parallel. This protocol mirrored the sample protocol, except that a commercial serum matrix spiked with synthetic oligonucleotides mimicking different concentrations of miRNA122 (20000, 5000, 1250, 313, 78, 20, and 5 pg/mL) was used instead of the patient serum sample. Serum matrix without spike-in was used as the blank sample.

### Statistical analysis

Statistical analyses were performed using GraphPad Prism (version 9.4.1, GraphPad Software, San Diego, CA, USA). Spearman's rank correlation coefficient (r) was used to assess the relationship between miRNA122 levels and traditional liver biomarkers (ALT, AST, and TBL), Spearman's rank correlation coefficient (r) was used. A p-value of less than 0.05 was considered statistically significant.

## Results

### Study cohort description

The study population comprised eight patients with ALI. To assess changes during disease progression, the patients were observed over three different visits from the time the acute episode was detected. The mean age was 50 years. The biochemical and demographic characteristics of the study population are shown in Table 1. Viral hepatitis and autoimmune

**Table 1. Demographics and clinical characteristics of the study cohort.**

| Parameter | Clinical data | | |
|---|---|---|---|
| **Age** | 50 ± 15 | | |
| **Sex (male/ female)** | 3/ 5 | | |
| **Type of injury (%)** | | | |
| Hepatocellular | 87.5% | | |
| Cholestatic | 12.5% | | |
| **Severity (%)** | | | |
| Mild | 25% | | |
| Moderate | 50% | | |
| Severe | 25% | | |
| **Biochemistry** | **Visit 1** | **Visit 2** | **Visit 3** |
| TBL mg/dL | 10.9 ± 10.3 | 9.7 ± 12.3 | 1.6 ± 1.7 |
| AST IU/L | 679 ± 240 | 453 ± 285 | 131 ± 163 |
| ALT IU/L | 824 ± 445 | 503 ± 312 | 110 ± 138 |
| ALP IU/L | 154 ± 87 | 142 ± 38 | 109 ± 52 |

hepatitis were ruled out in all patients through serology testing and/or clinical judgement. Additionally, there was also no clear temporal sequence, nor was there sufficient evidence from patient follow-up and medical records to support a convincing diagnosis of idiosyncratic drug-induced liver injury.

## DCL PCR-Free method performance

A seven-point calibration curve was constructed to quantify miRNA122 levels using the DCL PCR-Free method (S2 Fig). Each data point was obtained by spiking synthetic oligonucleotides mimicking miRNA122 into the serum matrix and performing duplicate measurements. Serum matrix that was not spiked-in was included as a blank (control). Sensitivity, precision, and accuracy were determined as described elsewhere [34]. The calculated precision and accuracy of the DCL PCR-Free method were 3.36% and 99.80%, respectively. The extrapolated sensitivity parameters, the limit of detection (LOD) and the lower limit of quantification (LLOQ), were 10.22 pg/mL and 19.53 pg/mL, respectively (S1 Table).

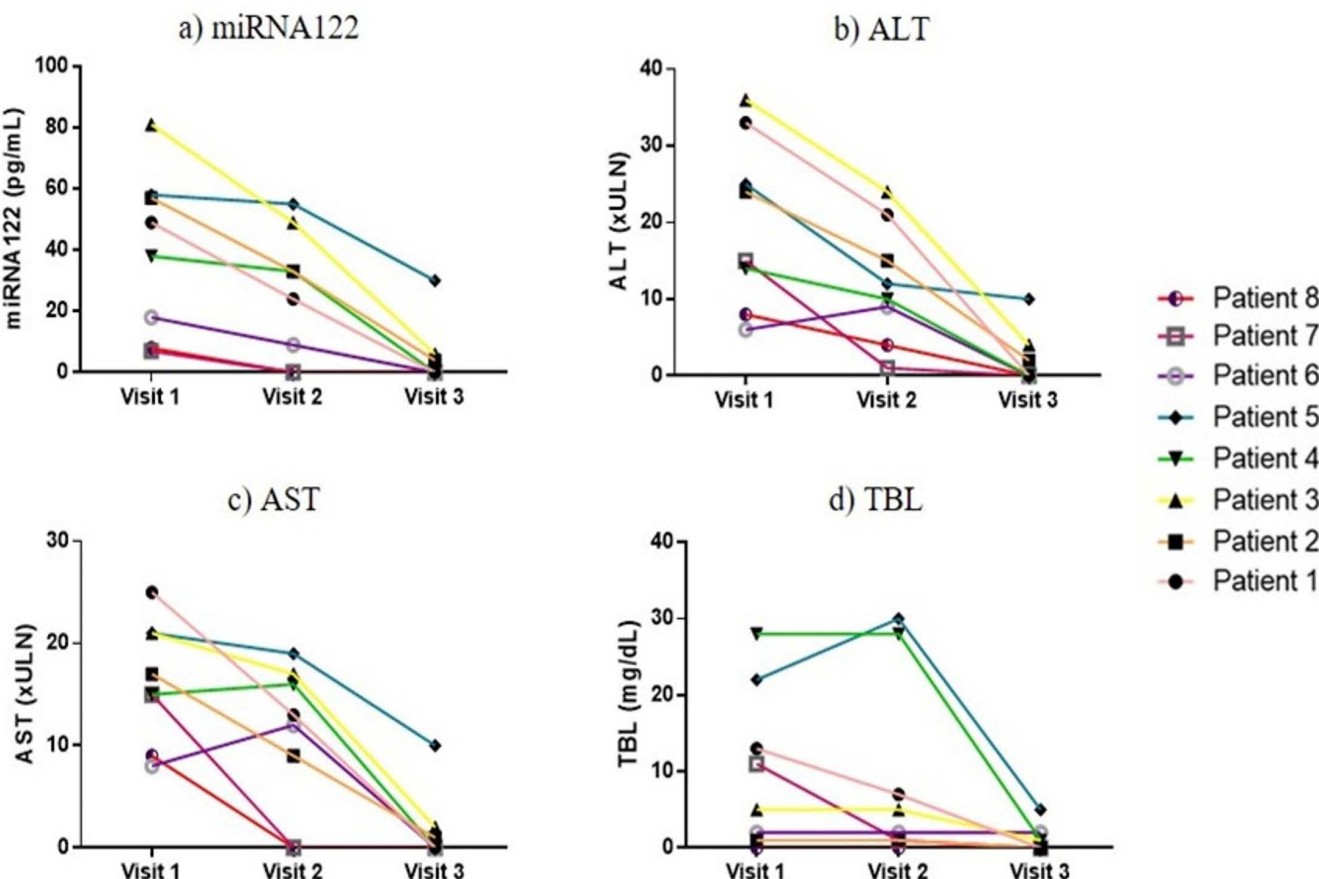

**Fig 1. Biomarker profiles over the course of the liver event:** a) microRNA122-5p (miRNA122); b) Alanine aminotransferase (ALT); c) Aspartate aminotransferase (AST); d) Total bilirubin (TBL). The visits were: visit 1 (day 1, recognition); visit 2 (day 7); and visit 3 (≥30 days after recognition of the acute episode).

### Profile of miRNA122 during the course of the liver event and comparison with standard biomarkers

For the quantitative analysis, the patient's miRNA122 data were quantified using the calibration curve shown in S2 Fig. Fig 1. shows the monitoring of the levels of miRNA122 compared to the standard biomarkers during visits 1, 2 and 3. All patients showed decreased in the levels of the three standard biomarkers and miRNA122 during the recovery process, particularly between visits 1 and 3.

Correlation analyses were performed between miRNA122 and the standard biomarkers using the Spearman correlation test (Fig 2 and S2 Table). The correlation coefficient between miRNA122 and ALT was 0.8150, indicating a strong positive linear relationship with high statistical significance (p < 0.0001). Similarly, a strong positive correlation was found between miRNA122 and AST (r = 0.7895, p = 0.0002), suggesting that changes in miRNA122 levels are associated with changes in AST levels. In contrast, no correlation was found between miRNA122 and TBL (r = 0.2646; p = 0.3406).

## Discussion

The need to validate new methods that simplify the detection of microRNAs is particularly relevant in the case of ALI. In particular, the miRNA122 plays a key role in maintaining liver homeostasis by participating in the regulation of processes such as differentiation, inflammation, lipid metabolism and apoptosis. However, the precise role of miRNA122 in liver disease remains unclear. Knock-down of miRNA122 in mice has been shown to promote the development of steatohepatitis, fibrosis and HCC [39]. In the case of DILI by N-acetyl-para-aminophenol (APAP), both *in vitro* and *in vivo* models have demonstrated that knocking down miRNA122 is protective against liver damage [15]. During hepatocyte damage or stress, miRNA122 is released into the bloodstream where it is relatively stable, enabling its detection as a non-invasive biomarker. This study demonstrated the effectiveness of the DCL PCR-Free method in quantifying miRNA122 in patient serum during recovery from ALI and provided valuable insight into the robustness of the DCL PCR-Free method, particularly in terms of sensitivity, precision, and accuracy. The sensitivity results yielded a LOD of 10.22 pg/mL, enabling direct detection of miRNA122 without the need for pre-extraction and amplification. Additionally, the observed precision is consistent with that of commercially available Luminex assays, typically yielding values of less than 10%. The accuracy percentage demonstrates the robust performance of the assay, with values approaching 100% indicating its effectiveness in detecting and quantifying miRNA122.

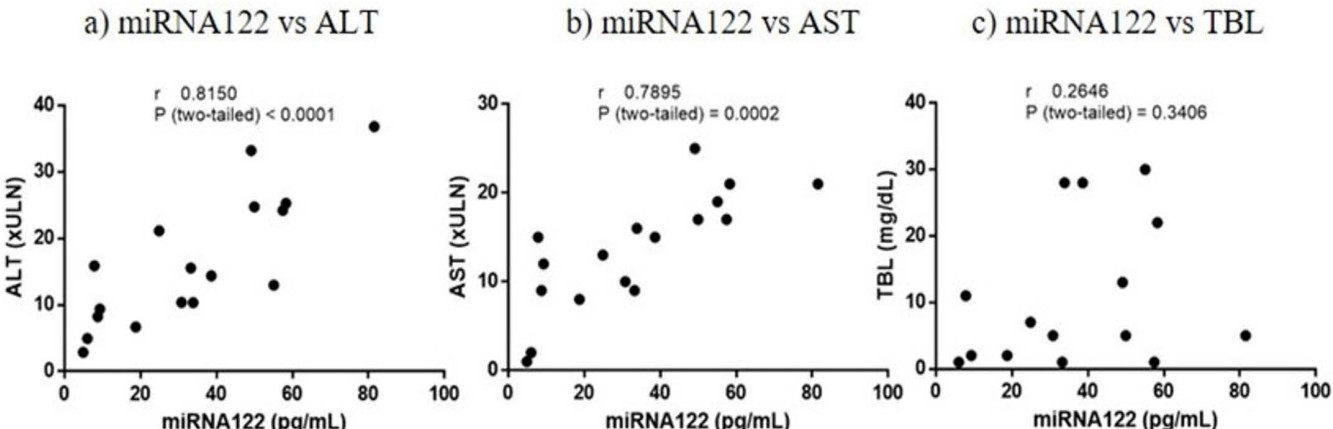

**Fig 2. Spearman's correlation analysis:** a) microRNA-122-5p (miRNA122) vs. Alanine Aminotransferase (ALT); b) miRNA122 vs. Aspartate Aminotransferase (AST); c) miRNA122 vs. Total Bilirubin (TBL). Abbreviation: xULN, times the upper limit of normal. Values below the lower limit of quantification (LLOQ) were not included in the analysis.

Our research further investigates the correlation between miRNA122 measurements obtained using the DCL PCR-Free method and traditional biomarkers assessed by conventional methods. We found that miRNA122 demonstrated a strong correlation with ALT (r = 0.8150) and AST (r = 0.7895) during the ALI episode, indicating a strong relationship between miRNA122 and both ALT and AST levels. These findings are consistent with previous analyses indicating the reliability of miRNA122 as an alternative to ALT and AST for calculating patterns of liver injury such as in DILI [40,41]. As expected, there was no correlation between miRNA122 and TBL, since the latter is a parameter of hepatic function. These results are consistent with recent research presented by Cueto-Sánchez et al. who studied a cohort of patients with ALI of various aetiologies, including DILI, autoimmune hepatitis and viral hepatitis [13].

The DCL PCR-Free method on Luminex xMAP platforms shows great promise for monitoring liver injury, as it can directly detect the candidate biomarker miRNA122 [5]. Unlike traditional biomarkers, miRNA122 offers several advantages, including its liver specificity and the potential to provide mechanistic insights into liver injury pathways during drug development and in daily clinical practice. By measuring levels of this microRNA directly, the DCL PCR-Free method enables clinicians to gain a deeper iunderstanding of the molecular processes underlying liver injury, allowing for more accurate and timely monitoring of treatment. Furthermore, using miRNA122 as a biomarker has the potential to enable the early detection of hepatotoxicity, even before symptoms or alterations in traditional biomarker profiles manifest. Studies have shown that miRNA122 levels are elevated in patients with established acetaminophen (paracetamol)-induced liver injury, even when current markers such as ALT are normal [42,43]. This is particularly important in clinical trials and drug development, where continuous monitoring of liver health is essential.

In conclusion, the use of the DCL PCR-Free method represents an advancement that could facilitate the early and accurate detection of miRNA122 and enable the monitoring of ALI during recovery. Future studies involving larger cohorts are needed to establish the broader research and clinical applicability of this methodology for using miRNA122 as a liver mechanistic biomarker.

## Supporting information

**S1 Fig. Cartoon illustrating the general workflow of DCL.** The DGL Probe captures the complementary single stranded nucleic acid (miRNA) sequence, forming the Chemical Pocket. Once the target miRNA is fully hybridized, the SMART-Base Biotin is incorporated and covalently linked to the backbone of the DGL Probe, resulting in the Chemical Lock up. The duplex is then detected using a reporter molecule, such as Streptavidin Phycoerythrin, which specifically recognizes the biotin tag. The final read-out is performed using the Luminex platform. (Anal. Methods, 2023,15, 6139–6149).
(JPG)

**S2 Fig. Calibration Curve.** Calibration curve was constructed by utilizing a 5PL non-linear regression model, where MFI Average values were plotted against the logarithm of phase 10 of concentration of synthetic oligonucleotide mimicking miRNA122. Seven-point concentrations were tested by spiking in commercially available serum matrix (MP Biomedicals™, Cat. Number 11465055). Concentrations were respectively 20000.00, 5000.00, 1250.00, 312.50., 78.13, 19.53 and 4.88 pg/mL. Non-spiked-in serum was used as negative control. Each measurement was conducted in duplicate.
(JPG)

**S1 Table. Technical specifications of DCL PCR-Free method.**
(PDF)

**S2 Table. Correlation test results.**
(PDF)

**S1 Text. Raw data used to generate Fig 1 and Fig 2, as well as S2 Fig and, S1 and S2 Tables.**
(PDF)

## Acknowledgments

The authors would like to thank Prof. James Dear (University of Edinburgh) for his invaluable scientific guidance on the miRNA122 biomarker. We would also like to thank Dr Christa Nöhammer from the AIT Austrian Institute of Technology for her insightful contributions on the Luminex xMAP technology. Finally, we would like to thank all the patients and healthy volunteers who participated in this study.

## Author contributions

**Conceptualization:** Antonio Marín-Romero, Daniel E. Di Zeo-Sánchez, Juan J. Díaz-Mochón, M. Isabel Lucena González, Raúl J. Andrade, Camilla Stephens, Salvatore Pernagallo.

**Formal analysis:** Antonio Marín-Romero, Mavys Tabraue-Chávez, Juan J. Díaz-Mochón, Salvatore Pernagallo.

**Funding acquisition:** Juan J. Díaz-Mochón, Salvatore Pernagallo.

**Investigation:** Antonio Marín-Romero, Mavys Tabraue-Chávez, Juan J. Díaz-Mochón.

**Methodology:** Antonio Marín-Romero, Mavys Tabraue-Chávez, Salvatore Pernagallo.

**Resources:** Marina Villanueva-Paz, Jose M. Pinazo-Bandera, Judith Sanabria-Cabrera, Miren García-Cortés, Juan J. Díaz-Mochón, Camilla Stephens, Salvatore Pernagallo.

**Supervision:** M. Isabel Lucena González, Raúl J. Andrade, Camilla Stephens, Salvatore Pernagallo.

**Validation:** Antonio Marín-Romero, Salvatore Pernagallo.

**Visualization:** Antonio Marín-Romero, Juan J. Díaz-Mochón, Salvatore Pernagallo.

**Writing – original draft:** Antonio Marín-Romero, Daniel E. Di Zeo-Sánchez, Salvatore Pernagallo.

**Writing – review & editing:** Antonio Marín-Romero, Daniel E. Di Zeo-Sánchez, Marina Villanueva-Paz, Juan J. Díaz-Mochón, Camilla Stephens, Salvatore Pernagallo.

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
