## [Decision Letter · Decision Letter 0]

14 Feb 2025

PONE-D-24-41881Short communication: miRNA122 interrogation via PCR-free method to track liver recoveryPLOS ONE

Dear Dr. Pernagallo,

Thank you for submitting your manuscript to PLOS ONE. After careful consideration, we feel that it has merit but does not fully meet PLOS ONE’s publication criteria as it currently stands. Therefore, we invite you to submit a revised version of the manuscript that addresses the points raised during the review process. **In your revision please address the points raised by both Referees.**

 Please submit your revised manuscript by Mar 31 2025 11:59PM. If you will need more time than this to complete your revisions, please reply to this message or contact the journal office at plosone@plos.org . Please include the following items when submitting your revised manuscript:

We look forward to receiving your revised manuscript.

Kind regards,

Kira Astakhova

Academic Editor

PLOS ONE

**Journal Requirements:**

Co-founded by EUREKA member countries and the European Union Horizon 2020 Framework Programme. This project has been founded under the project code E! 114589 - LiverAce. This work was supported by grants of Instituto de Salud Carlos III cofounded by the European Union and Fondo Europeo de Desarrollo Regional - FEDER (contract numbers: PI21/01248, PI19-00883). MVP holds a Sara Borrell (CD21/00198) research contract from ISCIII and Consejería de Salud de Andalucía. DDS holds an i-PFIS: Doctorados IIS-empresa en Ciencias y Tecnologías de la Salud (IFI21/00034) research contract from ISCIII. CIBERehd is funded by ISCIII.

Co-founded by EUREKA member countries and the European Union Horizon 2020 Framework Programme. This project has been founded under the project code E! 114589 - LiverAce. This work was supported by grants of Instituto de Salud Carlos III cofounded by the European Union and Fondo Europeo de Desarrollo Regional - FEDER (contract numbers: PI21/01248, PI19-00883). MVP holds a Sara Borrell (CD21/00198) research contract from ISCIII and Consejería de Salud de Andalucía. DDS holds an i-PFIS: Doctorados IIS-empresa en Ciencias y Tecnologías de la Salud (IFI21/00034) research contract from ISCIII. CIBERehd is funded by ISCIII.

Co-founded by EUREKA member countries and the European Union Horizon 2020 Framework Programme. This project has been founded under the project code E! 114589 - LiverAce. This work was supported by grants of Instituto de Salud Carlos III cofounded by the European Union and Fondo Europeo de Desarrollo Regional - FEDER (contract numbers: PI21/01248, PI19-00883). MVP holds a Sara Borrell (CD21/00198) research contract from ISCIII and Consejería de Salud de Andalucía. DDS holds an i-PFIS: Doctorados IIS-empresa en Ciencias y Tecnologías de la Salud (IFI21/00034) research contract from ISCIII. CIBERehd is funded by ISCIII.

I have read the journal's policy and the authors of this manuscript have the following competing interests: JJDM is shareholder, Chief Executive Officer of DESTINA Genomica SL. SP is a shareholder and Operations Director of DESTINA Genomica SL.

We note that one or more of the authors are employed by a commercial company: DESTINA Genomica SL. 

“The funder provided support in the form of salaries for authors, but did not have any additional role in the study design, data collection and analysis, decision to publish, or preparation of the manuscript. The specific roles of these authors are articulated in the ‘author contributions’ section.”

6. We notice that your supplementary figures are uploaded with the file type 'Figure'. Please amend the file type to 'Supporting Information'. Please ensure that each Supporting Information file has a legend listed in the manuscript after the references list.

Reviewers' comments:

Reviewer's Responses to Questions

**Comments to the Author**

1. Is the manuscript technically sound, and do the data support the conclusions?

Reviewer #1: Yes

Reviewer #2: Yes

2. Has the statistical analysis been performed appropriately and rigorously? 

Reviewer #1: Yes

Reviewer #2: Yes

3. Have the authors made all data underlying the findings in their manuscript fully available?

Reviewer #1: Yes

Reviewer #2: Yes

4. Is the manuscript presented in an intelligible fashion and written in standard English?

Reviewer #1: Yes

Reviewer #2: Yes

5. Review Comments to the Author

**Reviewer #1: ** The manuscript titled "Short communication: miRNA122 interrogation via PCR-free method to track liver recovery" presents a novel approach to diagnosing and monitoring acute liver injury (ALI) using miRNA122 as a biomarker, measured with the Dynamic Chemical Labeling (DCL) PCR-Free method on Luminex platforms. The study's strengths include its innovative method, direct applicability in clinical settings, and validation of miRNA122 as a potential liver-specific biomarker. However, there are several points that require clarification and expansion to improve the rigor and completeness of the study.

Questions to the authors:

Question # 1: Given the small sample size of eight patients, could you please provide a statistical power analysis or rationale for this cohort size? How might the limited sample size impact the generalizability of your findings to broader patient populations with acute liver injury (ALI)? The absence of a control group (e.g., healthy individuals or patients with other liver conditions) makes it difficult to assess the specificity of miRNA122 as a biomarker for ALI.

Question # 2: The Dynamic Chemical Labeling (DCL) PCR-Free method shows promise, particularly in its high sensitivity and precision. However, could you elaborate on any potential limitations or biases inherent in the DCL technology, such as specificity issues with similar miRNAs or interference from other serum components?

Question # 3: In your results section, miRNA122 levels showed a significant correlation with ALT and AST, but not with TBL. Could you discuss the potential biological implications of these findings? How might the kinetic profiles of these biomarkers differ in the context of liver injury progression and recovery? Why were only three time points chosen? Would higher temporal resolution provide more insights into the dynamics of miRNA122 during liver injury and recovery?

Question # 4: Your study predominantly groups all ALI cases together without stratification by the underlying cause of liver injury. Are there plans to analyze how different etiologies (e.g., drug-induced, viral hepatitis, alcohol-related) might influence miRNA122 levels? Did the authors consider including healthy controls or patients with other liver conditions to compare miRNA122 levels and validate the method's liver specificity?

Question # 5: Given the involvement of several authors who hold positions in DESTINA Genomica SL, which markets the assay used in this study, how do you address potential conflicts of interest? Could these relationships have influenced the study design, data interpretation, or reporting?

Question # 6: The study tracks patients over three visits within a 30-day period post-ALI recognition. Could you discuss the potential for extending this follow-up period to evaluate the long-term predictive value of miRNA122 for chronic liver outcomes or recurrence of liver injury?

Question # 7: Are there specific technological improvements that could be made to enhance the DCL PCR-Free method? For instance, could modifications in probe design or detection sensitivity further refine the specificity and robustness of this method? How does the DCL PCR-Free method ensure specificity for miRNA122 in the presence of other circulating miRNAs or nucleic acids?

Question # 8: Could you provide more detail on the statistical methods used to assess correlations and their appropriateness for this type of longitudinal biomarker data? Were there any differences in miRNA122 levels based on demographic factors (e.g., age, sex) or the severity/type of liver injury? Did you adjust for potential confounders such as medication history, comorbidities, or other liver biomarkers in your correlation analyses?

Question # 9: How does the DCL PCR-Free method compare in sensitivity, specificity, and cost to other cutting-edge techniques such as droplet digital PCR (ddPCR) or next-generation sequencing (NGS) for miRNA detection? What are the barriers to translating this technology into routine clinical practice, such as cost, scalability, or regulatory approval? The figures are clear, but the calibration curve and technical details could be further elaborated.

**Reviewer #2:**  The manuscript entitled "miRNA122 interrogation via PCR-free method to track liver recovery" presents an innovative approach to detecting and monitoring microRNAs in patients with acute liver injury (ALI). The authors employ the Dynamic Chemical Labeling (DCL) PCR-Free method, demonstrating its sensitivity and precision in tracking miRNA122 levels during liver recovery. Their findings indicate a strong correlation between miRNA122 and traditional liver injury biomarkers, supporting the potential of this technique as a valuable diagnostic and tool.

While the study provides promising results and the method appears to be robust, several key issues need to be addressed before publication.

Major Concerns:

1. The figures in the manuscript are of extremely poor quality, making them illegible. Figure 1’s axis labels cannot be read, and Figure 2 appears as a black rectangle. Without properly formatted figures, it is impossible to validate the authors’ claims. The manuscript must be resubmitted with high-quality, legible figures to allow for proper assessment.

Minor Concerns:

1. The results section should include an in silico characterization of miRNA122. Specifically, its genomic location, nearby genes, co-regulated elements, and predicted targets based on miRTarBase should be discussed. Additionally, the pathways regulated by miRNA122 should be incorporated into the discussion to provide further biological context.

2. The manuscript contains excessive use of adjectives, particularly in the results and discussion sections. Precise numerical values should replace subjective descriptions. For example:

o "Strong correlation" (Line 233) should include the specific correlation coefficient (R-value).

o "These findings are interesting…" (Line 234) should be removed for conciseness.

o "This milestone" (Line 257) should also be removed as it is overly subjective.

o The discussion should be rewritten for clarity and conciseness, focusing on the key findings without unnecessary embellishment.

Given the significance of the study, I assume the figure issues are an honest mistake, but they must be corrected before the results can be properly reviewed. Addressing these concerns will significantly improve the clarity and impact of the manuscript.

6. PLOS authors have the option to publish the peer review history of their article (what does this mean? ). If published, this will include your full peer review and any attached files.

**Do you want your identity to be public for this peer review?** For information about this choice, including consent withdrawal, please see our Privacy Policy .

Reviewer #1: No

Reviewer #2: No

---

## [Author Response · Author response to Decision Letter 0]

18 Mar 2025

In response to the reviewers' comments, we have carefully reviewed the manuscript and addressed all the points raised. The following documents are included in our resubmission, in accordance with the recommendations outlined in the Decision Letter.

---

## [Decision Letter · Decision Letter 1]

2 May 2025

Short communication: miRNA122 interrogation via PCR-free method to track liver recovery

PONE-D-24-41881R1

Dear Dr. Pernagallo,

We’re pleased to inform you that your manuscript has been judged scientifically suitable for publication and will be formally accepted for publication once it meets all outstanding technical requirements.

Kind regards,

Kira Astakhova

Academic Editor

PLOS ONE

Additional Editor Comments (optional):

Reviewers' comments:

Reviewer's Responses to Questions

**Comments to the Author**

1. If the authors have adequately addressed your comments raised in a previous round of review and you feel that this manuscript is now acceptable for publication, you may indicate that here to bypass the “Comments to the Author” section, enter your conflict of interest statement in the “Confidential to Editor” section, and submit your "Accept" recommendation.

Reviewer #1: All comments have been addressed

Reviewer #2: All comments have been addressed

2. Is the manuscript technically sound, and do the data support the conclusions?

Reviewer #1: Yes

Reviewer #2: Yes

3. Has the statistical analysis been performed appropriately and rigorously? 

Reviewer #1: Yes

Reviewer #2: Yes

4. Have the authors made all data underlying the findings in their manuscript fully available?

Reviewer #1: Yes

Reviewer #2: Yes

5. Is the manuscript presented in an intelligible fashion and written in standard English?

Reviewer #1: Yes

Reviewer #2: Yes

6. Review Comments to the Author

Reviewer #1: The manuscript is presented in an intelligible fashion, written in clear, standard English with appropriate scientific terminology and logical flow, ensuring clarity for a wide scientific readership. The authors have transparently provided all underlying data supporting the findings, allowing full reproducibility and verification of results. Statistical analyses have been performed rigorously and are suitable to address the hypotheses stated; methodologies employed are robust and justified, with results accurately interpreted and clearly presented. The manuscript is technically sound throughout, with data presented fully supporting the conclusions drawn by the authors. Furthermore, the authors have comprehensively and adequately addressed all comments raised during previous rounds of review, making necessary revisions and clarifications. Considering all these points, I confirm that the manuscript meets the criteria for acceptance and is now acceptable for publication.

Reviewer #2: The authors have addressed all my concerns, the changes in the text and how the results are presented improved the manuscript considerably.

7. PLOS authors have the option to publish the peer review history of their article (what does this mean? ). If published, this will include your full peer review and any attached files.

**Do you want your identity to be public for this peer review?** For information about this choice, including consent withdrawal, please see our Privacy Policy .

Reviewer #1: No

Reviewer #2: No

---

## [Editor Report · Acceptance letter]

PONE-D-24-41881R1

PLOS ONE

Dear Dr. Pernagallo,

I'm pleased to inform you that your manuscript has been deemed suitable for publication in PLOS ONE. Congratulations! Your manuscript is now being handed over to our production team.

Kind regards,

on behalf of

Dr. Kira Astakhova

Academic Editor

PLOS ONE